# Preparation of Hollow Porous Carbon Nanofibers and Their Performance and Mechanism of Broadband Microwave Absorption

**DOI:** 10.3390/ma15207273

**Published:** 2022-10-18

**Authors:** Rui Shao, Fang Wang, Shenglin Yang, Junhong Jin, Guang Li

**Affiliations:** 1College of Materials Science and Engineering, Donghua University, Shanghai 201620, China; 2Sinopec Yizheng Chemical Fibre Company Limited, Yangzhou 225000, China

**Keywords:** hollow porous carbon nanofiber (HPCNF), composite, microwave absorption, reflection loss

## Abstract

Developing microwave absorbing composites with lightweight and wide absorption bands is an essential direction for electromagnetic wave stealth and shielding application. In this article, PAN/PMMA blend fibers and sheath-core blend fibers with PAN/PMMA as the sheath and PMMA as the core were spun by uniaxial and coaxial electrostatic spinning, respectively. Porous carbon nanofiber (PCNF) and hollow porous carbon nanofiber (HPCNF) were obtained after pre-oxidation and carbonization of the corresponding two precursor fibers. The microwave absorption composite samples with PCNF and HPCNF as absorbents and paraffin as matrix were prepared, respectively. Their electromagnetic parameters were investigated by the reflective-transmission network parameter method. The microwave absorption properties of the corresponding composites were calculated based on a model for a single-layer planewave absorber from electromagnetic parameters. The results showed diversity between the microwave absorbing performance of the composites filled with PCNF and HPCNF. HPCNF performs better than PCNF as an absorbent; that is, the lowest reflection loss of composite filled with HPCNF is −20.26 dB and the effective bandwidth (lower than −10 dB) is to 4.56 GHz, while the lowest reflection loss of a composite filled with PCNF is −13.70 dB, and the effective bandwidth (lower than −10 dB) is 2.68 GHz when the absorbent content is 7%, and the thickness is 3 mm. Much lower reflection loss and a wider absorption band could be expected from HPCNF. The presence of a hollow structure in HPCNF, which may increase the degree of polarization and provide more interfaces for the interference phase extinction of reflected electromagnetic waves, might help to improve the attenuation of electromagnetic waves and broaden the absorption band.

## 1. Introduction

With the rapid development of information technology, the increasingly refined positioning and tracking technologies for military equipment have significantly changed the pattern of military offense and defense. In order to improve the survivability and combat capability of weapons, it is urgent to develop advanced electromagnetic wave stealth technology. “Stealth” is not the disappearance of an object but the use of absorbing materials to absorb and attenuate electromagnetic waves [1]. Therefore, microwave absorbing materials are the basis of arms and equipment stealth.

Microwave absorbing materials are characterized by the ability to absorb incident electromagnetic waves and dissipate them through dielectric loss, magnetic loss, interference loss, and other forms. According to the principle of wave absorption, absorbing materials can be divided into two types of absorption and interference [2,3]. Absorbing-type materials operate through utilizing the electromagnetic properties of the material itself, converting the incident electromagnetic waves inside the materials into other energy consumption according to a specific electromagnetic loss mechanism, while interfering-type materials take advantage of electromagnetic waves in the round-trip free space and coherent waves generated between different interfaces of the materials. When the thickness of the two interfaces meets a multiple of 1/4 wavelength, interference phase elimination occurs, reducing the reflectivity of electromagnetic waves.

Absorbing-type materials mainly include magnetic and dielectric wave absorbers based on electromagnetic wave absorption mechanism. Ferrite and other magnetic nanoparticles belong to magnetic absorbers, which dissipate electromagnetic waves through hysteresis loss, eddy current loss, and ferromagnetic resonance. Dielectric absorbers are represented by carbon materials such as carbon black, carbon fiber, carbon nanotube, etc., [4,5,6] and are dominated by mechanisms such as resistive loss and dielectric relaxation. Despite a strong ability to absorb electromagnetic waves [7], the magnetic absorbers suffer from high density and poor absorption in the low-frequency range [8,9]. Thus, magnetic absorbers are usually utilized in conjunction with carbon materials. For example, Gao [10] et al. used the mixture of carbon nanotube (CNT) and Fe_3_O_4_ as the hybrid absorbent to prepare CNT-Fe_3_O_4_/epoxy composites. When the hybrid absorbent is added up to 30%, the composite with the thickness of 2.5 mm presented good microwave absorption performance, the frequency bandwidth below −10 dB reaching 7.2 GHz. The preparation of dendritic Co nanoparticles [11], dendritic FeCo, and rose-shaped CoNi has also been studied as an absorbent [12]. More and more studies have found that morphological modulation of carbon materials can improve microwave absorption performance. For example, when porous carbon fiber [13] is used as an absorber, it shows better performance than carbon fiber with the same diameters. The porous spiral carbon fiber prepared by Cao et al. [14] has a high specific surface area and high pore volume due to the existence of a spiral and porous structure when it is used as an absorbent with 10% addition; the effective band can be broad to 4.8 GHz with a thickness of 1.87 mm. To further reduce the weight of absorbents, hollow carbon sphere and hollow carbon fiber [15,16,17,18] have received attention as microwave absorbents. Li [19] et al. prepared hollow carbon sphere and used it as an absorbent with 20% addition; the lowest reflection loss reached −23 dB with an effective bandwidth of 4.4 GHz at a thickness of 1.5 mm. Hence, the existence of pore structure, spiral structure, and hollow structure can all increase the specific surface area to improve the microwave absorption performance of the material.

Based on the previous studies of the group, it was demonstrated that the presence of pores in carbon fibers could increase the transmission path and interference probability of electromagnetic waves and enhance microwave absorption based on both absorbing and interring mechanism, the absorption broadband reaching 2.0 GHz [13,20]. On the basis of pores, the presence of a hollow structure can increase the specific surface area of the material, and the increase of interface makes the possibility of an interference phase extinction larger and at the same time reduces the mass of the material. Therefore, hollow porous carbon nanofiber with a hollow structure was presented as a highly effective absorbent. The effects of hollow and porous structure on their electromagnetic wave attenuation mechanism and absorbing frequency band are investigated in this paper, laying the groundwork for realizing the objective of “light, thin, wide, and strong” absorbing composites.

## 2. Materials and Methods

### 2.1. Materials

Polyacrylonitrile (PAN) (Shanghai Petrochemical Co., Ltd., Shanghai, China); Polymethyl methacrylate (PMMA) (Fengyuan Plastic Chemical Co., Ltd., Nanchang, Jiangxi Province, China); N,N-dimethylformamide (DMF) (Sinopharm, Bejing, China); Porous carbon fiber (PCF), prepared by wet spinning and following carbonization in our laboratory with the average fiber diameter of 40 micrometer under a carbonization temperature of 1000 °C.

### 2.2. Preparation of Porous Carbon Nanofiber (PCNF)

PAN was used as the carbon precursor and PMMA as the pore-forming component. PAN and PMMA were mixed and dissolved in DMF in a ratio of 3:1 when the solid content was controlled at 17%. The obtained homogeneous solution was used for electrospinning to prepare a PAN/PMMA blend fiber with a diameter in the range of 400–800 nm. The electrospinning process parameters included using an 18-gauge needle, a pushing rate of 0.9 mL/h, a receiving distance of 15 cm, and 18 kV voltage.

We put an appropriate amount of PAN/PMMA blend fiber in a muffle furnace for pre-oxidation, with a temperature rise rate of 1 °C/min to 250 °C, held it for 1 h, and then allowed it to cool naturally to room temperature. Then, pre-oxidized PAN/PMMA fiber was carbonized in a tube furnace, using N_2_ as the protective gas, with a temperature rise rate of 2 °C/min to 1000 °C, holding for 2 h, and then cooling to room temperature. PCNF was obtained. 

### 2.3. Preparation of Hollow Porous Carbon Nanofiber (HPCNF)

The sheath-core PAN/PMMA fibers were obtained by coaxial electrospinning, in which the sheath layer was the same as that of the PCNF, and the core layer was a homogeneous solution of PMMA with a solid content of 20%. The electrospinning parameters were as follows: needle type: cortex with 18-gauge, core with 14-gauge; push rate: sheath with 0.5 mL/h, core part with 0.3 mL/h; receiving distance of 15 cm; 18 kV voltage.

The pre-oxidation and carbonization treatment were kept the same as mentioned above. PMMA in the core layer was decomposed completely at high temperature to form a central through hole, and the PMMA in the sheath layer was also decomposed to form holes to obtain HPCNF.

### 2.4. Preparation of Coaxial Ring Samples for Microwave Absorbing Composites

The electromagnetic parameters of the microwave absorbing composites were measured by preparing coaxial ring samples where PCNF and HPCNF were used as absorbent, respectively. The prepared PCNF and HPCNF were ground firstly in a mortar till to micrometer scale in length, then mixed with paraffin according to the absorbent content of 5%, 7%, and 9%, respectively, until uniformly dispersed. We placed the mixture in a ring-shaped mould after pressing on a tablet machine using pressure of 5 Mpa for 5 min, and obtained a coaxial ring sample. The coaxial ring sample size: inner diameter: 3.04 mm, outer diameter: 7 mm, thickness: 2 mm.

### 2.5. Measurement and Characterization

#### 2.5.1. Characterization of Morphology

The morphology of PCNF, HPCNF, and the coaxial ring of microwave absorbing composites was observed using a cold field emission scanning electron microscopy (SEM, SU8010, Hitachi, Ltd., Tokyo, Japan), at a voltage of 3 kV and 1 kV, respectively. Gold was sprayed on the sample surface before observation.

#### 2.5.2. Characterization of XPS Spectrum

The structure of HPCNF was analysed using the X-ray photoelectron spectrometer (Escalab 250Xi, Thermo Fisher Scientific, Waltham, MA, USA) with the test elements C, N, and O.

#### 2.5.3. Characterization of Raman Spectroscopy

Laser Raman spectroscopy (inVia-Reflex, Renishaw Corporation, London, United Kingdom) was used to test the graphitisation of carbon fibers in the wavelength range 500–2500 cm^−1^.

#### 2.5.4. Characterization of the Electromagnetic Parameters

The electromagnetic parameters of the prepared coaxial ring samples filled with PCNF and HPCNF, respectively, were measured by the reflection–transmission network parameter method using Agilent’s HP8722ES vector network (Agilent Corporation, Santa Clara, CA, USA) analyzer in the frequency range of 2–18 GHz.

## 3. Results and Discussion

### 3.1. The Morphology of PCNF and HPCNF and Their Dispersion in Paraffin 

The scanning electron microscope images of PCNF are shown in Figure 1a,b. The fiber diameter is relatively uniform, in the range of 400–800 nm, there are many pores on the cross section, and the pore size is 5–50 nm.

The morphological structure of HPCNF is shown in Figure 1c,d. It seems the prepared HPCNF is relatively uniform, with diameters ranging from 600–800 nm, and demonstrates that the prepared fibers have both hollow and porous structures with a hollowness of about 30–50%. The specific surface area is in the range of 120–150 m^2^.g^−1^, depending on the diameter of the fibers.

The morphology of the carbon fiber/paraffin coaxial ring sample is shown in Figure 1e. It can be seen that the fiber and paraffin are well mixed and more uniformly adhered to the paraffin, and the carbon fiber keeps the fiber morphology with a short rod shape.

### 3.2. Analysis of the HPCNF

Figure 2 shows the XPS spectrum of the HPCNF and the Raman spectroscopy of the HPCNF and PCNF. Based on Figure 2a, it can be seen that the HPCNF contains mainly three elements, namely C, N, and O, including 87.49% of C, 7.39% of N, and 5.12% of O. Figure 2b shows the high-resolution C 1s spectrum of HPCNF, which represents the highest proportion of C=C functional groups, indicating the high degree of graphitization of HPCNF. Figure 2c shows the high-resolution N 1s spectrum of HPCNF with four fitted peaks, namely pyridinic N at 398.0 eV, pyrrolic N at 399.7 eV, graphitic N at 401.0 eV, and oxidized N at 402.1 eV [21].

From Figure 2d, both HPCNF and PCNF show G and D peaks at 1350 cm^−1^ and 1580 cm^−1^. For carbon materials, the D-peak represents disordered vibrations caused by defects in the material, namely amorphous carbon; the G-peak represents in-plane stretching vibrations of sp^2^ hybridisation caused by the lamellar structure inside the fibers. The higher the intensity of the G-peak, the more regular the arrangement of the graphite layer and the higher the degree of graphitization. The R parameter, R = I_D_/I_G_, is usually used to indicate the degree of graphitization of the carbon material. The R value for PCNF and HPCNF is 0.989 and 0.993, respectively, as both have the same carbonisation temperature, and their R parameter is similar. As a microwave absorbent, a high R value may lead to high conductivity, which is not better for impedance matching.

### 3.3. Microwave Absorption Performance

Since both paraffin wax and resin do not affect electromagnetic waves, it is feasible to use paraffin wax instead of resin to prepare composite samples. The complex permittivity (ε) and complex permeability (μ) of the composites containing PCNF and HPCNF were measured by the reflection–transmission network parameter method, which represents the dielectric loss ability and magnetic loss ability of the composites. Since the carbon absorbents are not magnetic, the magnetic loss capacity is almost zero, which means that μ′ = 1 and μ″ = 0.

#### 3.3.1. Analysis of Microwave Absorption Performance of PCNF

In order to explore the effects of PCNF content on the microwave absorption, PCNF/paraffin composites with PCNF loading of 5%, 7%, and 9% were prepared. The electromagnetic parameters of these composite samples are shown in Figure 3, where Figure 3a,b show the real and imaginary parts of the complex permittivity. With the increase of frequency, the change of ε′ and ε″ is small, but ε′ and ε″ increase greatly with the PCNF content. Moreover, it can be seen from Figure 3c that the dielectric loss factor tan δ_ε_ increases with the increase of PCNF content, indicating that the loss ability of the composites to electromagnetic waves keeps increasing when the PCNF content grows from 5% to 7%, and 9%.

According to the transmission line theory, the electromagnetic parameters obtained by the test can be used to calculate the reflection loss of the single-layer composite laminate to the electromagnetic wave through the following Equations (1) and (2): (1)$$Z=Z_{0}\sqrt{\frac{\mu_{r}}{\varepsilon_{r}}}tan\left[j\left(\frac{2\pi fd}{c}\right)\sqrt{\mu_{r}\varepsilon_{r}}\right]\;\;$$
(2)$$RL=-20\log_{10}\left|\frac{Z-Z_{0}}{Z+Z_{0}}\right|$$
where *RL* represents the reflection loss of the absorbing material to the electromagnetic wave, *Z* represents the characteristic impedance of the absorbing material, *Z*_0_ represents the impedance value in free space, *d* represents the thickness of the absorbing material, *ε_r_* and *μ_r_* represent the complex permittivity and complex permeability, respectively, *c* represents the speed of light, and *f* represents the frequency.

Taking the formula as the theoretical basis and using MATLAB software (MathWorks Corporation, R2016a, Neddick, MA, USA) for programming, the reflection loss of the prepared composites can be calculated through electromagnetic parameters, and then the parameters can be optimized to achieve the best microwave absorption performance of the composite. Based on the complex permittivity in Figure 3, the reflection loss of single-layer composite material with different PCNF contents and thicknesses was calculated as shown in Figure 4.

From Figure 4a, it can be seen that when the thickness is 3 mm and PCNF content is 5%, the minimum reflection loss of the composite is only −7.37 dB, the corresponding absorption peak frequency is 13.44 GHz, and the effective bandwidth (lower than −10 dB) is 0; when PCNF loading increases to 7%, the minimum reflection loss is −13.70 dB, the absorption peak frequency is 10.72 GHz, the effective frequency less than −10 dB is 2.68 GHz, and the absorption performance becomes better; when the PCNF content is 9%, the lowest reflection loss is −31.20 dB, the corresponding absorption peak frequency is 8.40 GHz, and the effective band is further widened to 2.72 GHz. Thus, the PCNF loading of 9% is the optimal among the three samples and shows the best performance of microwave absorption.

From the results in Figure 4b, it can be seen that as the thickness of the composite increases, the frequency corresponding to its lowest reflection shifts to lower frequencies. This result can be explained by the quarter-wavelength extinction method. Namely, the relationship between the thickness of the interference phase elimination of electromagnetic waves reflected by two interfaces and the electromagnetic wavelength should meet the following Equation (3).
(3)$$d_{m}=\frac{n\lambda }{4}=nc/\left(4f_{m}{\left(\left|\mu_{r}\right|\left|\varepsilon_{r}\right|\right)}^{\frac{1}{2}}\right)$$
where *d_m_* is the thickness, *n* = 1, 3, 5,...... *ε_r_* is complex permittivity relative, *μ_r_* is complex permeability, *λ* is the wavelength, *c* is the speed of light, and *f_m_* is the frequency. According to the above equation, the corresponding thickness of the composite where the interference occurs is inversely proportional to the absorption frequency. In other words, the larger the thickness of the composite, the smaller the absorption frequency corresponding to the interference will be.

#### 3.3.2. Analysis of Microwave Absorption Performance of HPCNF

In order to further utilize the pore structure in carbon fibers to dissipate electromagnetic waves and reduce the mass of the absorbing composites, we use HPCNF as a wave absorbent to investigate the contribution of a hollow structure to microwave dissipation. The electromagnetic parameters of the composite samples with different contents of HPCNF are shown in Figure 5, where the real and imaginary parts of the complex permittivity is given. With the increasing of frequency, the change of ε′ and ε″ is small, but ε′ and ε″ increase as the absorbent content keeps increasing. Similarly, consistent with PCNF absorbent, with the increasing content of HPCNF, the loss factor tan δ_ε_ increases gradually, as given in Figure 5c.

The reflection loss of the composites with different contents of HPCNF and at different thicknesses calculated based on the above complex permittivity are shown in Figure 6a,b.

According to Figure 6a, it can be seen that at a thickness of 3 mm, when the content of HPCNF is 5%, the lowest reflection loss is only −8.78 dB, which does not meet the requirement. While HPCNF content is increased to 7%, the lowest reflection loss reaches −20.26 dB, the absorption peak frequency is 9.20 GHz, and the effective bandwidth is widened to 4.56 GHz. When HPCNF loading is 9%, the lowest reflection loss is −19.02 dB at 8.40 GHz, and the effective bandwidth is 3.28 GHz. Among the three contents of HPCNF as an absorbent, the loading of 7% is optimal. As shown in Figure 6b, the frequency corresponding to the lowest reflection shifts to lower frequencies as the thickness of the composite increases, consistent with the results for PCNF as the absorbent, which can also be explained by the quarter-wavelength extinction method.

In order to find the absorption performance of HPCNF composites more intuitively, a three-dimensional graph of frequency, thickness, and reflection loss was made, as shown in Figure 6c. From the simulation results, it can be seen that when the content of HPCNF is 7%, and the thickness is 3 mm, the bandwidth with reflection loss less than −10 dB is 4.56 GHz and the lowest reflection loss is −20.26 dB, which presents excellent absorption performance and is better than that when PCNF acts as an absorbent with the same content. 

### 3.4. The Broadband Absorbing Principle of HPCNF

To further understand the influence of carbon fiber scale and morphology on microwave absorption performance, we plotted the reflection loss microwave of the composites made from three different scales and morphologies of carbon fibers as absorbents, respectively, together for comparison, as shown in Figure 7a. Among them, PCF, as shown in Figure 7b, is a large porous carbon fiber with an average diameter of 40 microns obtained by wet spinning and following carbonization. When the absorbent content is 7% and the thickness is 3 mm, the lowest reflection loss of the PCF composite is only −6.90 dB, and the absorption performance is poor. The lowest reflection loss of PCNF composite is −13.70 dB, and the effective bandwidth for reflection loss less than −10 dB is 2.68 GHz, so the absorption performance is improved compared to PCF. When HPCNF is used as an absorbent, the lowest reflection loss of the composite material is −20.26 dB, and the effective bandwidth increases to 4.56 GHz, with the best absorption performance.

The differences in the absorption performance of the above three carbon absorbents can be understood in terms of absorption and interference mechanisms [22,23]. As solid carbon materials can dissipate electromagnetic waves through resistive losses and dielectric polarization [24], PCNF and HPCNF with small diameters (400–800 nm) have a larger specific surface area and a more significant number of mesopores present (equivalent to defects) than PCF [13] with coarse diameters (40 µm), and under the excitation of electromagnetic field, PCNF and HPCNF undergo more and a greater degree of interfacial and defect-induced polarization, thus dissipating more electromagnetic waves than PCF. On the other hand, pores and hollow structure provide more interfaces to increase the transmission path of electromagnetic waves, as shown in Figure 8, increasing the probability of meeting the interference phase elimination [25] of reflected electromagnetic waves from different interfaces. In particular, the existence of a hollow structure provides more d values compared to the same thickness of a composite. For example, within the composite of thickness d, there are other interfaces of thickness d_1_ and d_2_. The corresponding frequency of electromagnetic waves to meet Equation (3) will also occur at interference phase elimination and with d located at different frequencies, thus broadening the frequency bandwidth of microwave absorption. As expected and shown in Figure 7, the absorption band of HPCNF composites is broader than that of PCNF composites at the same thickness. Furthermore, because of the existence of the hollow structure, the mass of the composites is reduced to a certain extent, which is in line with the direction of “lighter and wider” for microwave absorbing composites.

The results mentioned above can also be explained by the Debye relaxation mechanism. The Cole–Cole curves of the three absorbing composites were made by the complex dielectric constants (Figure 9). From the figures, it can be seen that all samples have linear and semicircular parts, indicating that the Debye relaxation process and resistance loss occur simultaneously when the electromagnetic waves enter the interior of the composites. With the growth of specific surface area and the appearance of hollow structure, the number of Cole–Cole semicircles increases and becomes larger, indicating that the polarization center increases, the probability of dipole polarization and interface polarization becomes larger, the polarization relaxation process improves, and the ratio of Debye relaxation process to dielectric loss is upgraded. It is consistent with the previous qualitative descriptions of the microwave absorption mechanisms of PCF, PCNF, and HPCNF composites.

In addition, the attenuation coefficient can reflect the absorbing ability of the composites comprehensively, and the Equation (4) is shown as follows.
(4)$$\alpha =\frac{\sqrt{2\pi f}}{c}\times \sqrt{\left(\mu^{\prime \prime }\varepsilon^{\prime \prime }-\mu^{\prime }\varepsilon^{\prime }\right)+\sqrt{{\left(\mu^{\prime \prime }\varepsilon^{\prime \prime }-\mu^{\prime }\varepsilon^{\prime }\right)}^{2}+{\left(\mu^{\prime }\varepsilon^{\prime \prime }+\mu^{\prime \prime }\varepsilon^{\prime }\right)}^{2}}}\;\;$$
where *α* represents the attenuation coefficient of the absorbing material, *ε*′ and *ε*″ represent the real and imaginary parts of the complex permittivity, *μ*′ and *μ*″ represent the real and imaginary parts of complex permeability, *c* represents the speed of light, and *f* represents the frequency.

The attenuation coefficients of absorbing composites with different absorbents are shown in Figure 10. The attenuation coefficient for the HPCNF composite is the highest among the three, followed by PCNF, while the attenuation coefficient for the PCF composite is the lowest. It is compatible with the change of microwave absorption performance of corresponding composites; that is, a larger attenuation coefficient corresponds to a stronger wave absorption performance and has a lower reflection loss of electromagnetic wave.

## 4. Conclusions

The sheath-core blend fiber was prepared by the coaxial electrospinning with PAN/PMMA as the sheath layer and PMMA as the core layer. After pre-oxidation and carbonization of the sheath-core blend fiber, HPCNF was obtained. The composites filled using HPCNF as an absorbent showed excellent microwave absorption, where the minimum reflection loss reached −20.26 dB, and the effective bandwidth (lower than −10 dB) was 4.56 GHz when HPCNF loading was 7% with a thickness of 3 mm. While the same content of PCNF was used as the absorber, the effective bandwidth was only 2.68 GHz. This indicates that the existence of holes and hollow structures increases the polarization and provides more interfaces that satisfy the interference cancellation of reflected electromagnetic waves, thereby improving the attenuation of electromagnetic waves and broadening the absorbing frequency band. HPCNF is an excellent lightweight, broadband absorbent.

## Figures and Tables

**Figure 1 materials-15-07273-f001:**
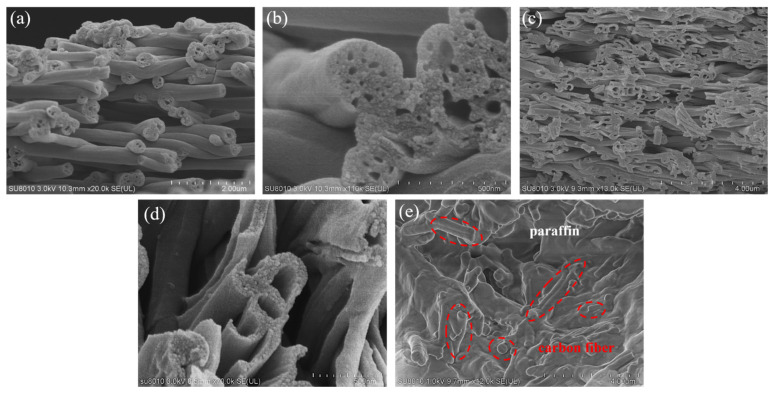
(**a**,**b**) Morphology of PCNF; (**c**,**d**) morphology of HPCNF; (**e**) morphology of the coaxial ring (red circles represent carbon fibers).

**Figure 2 materials-15-07273-f002:**
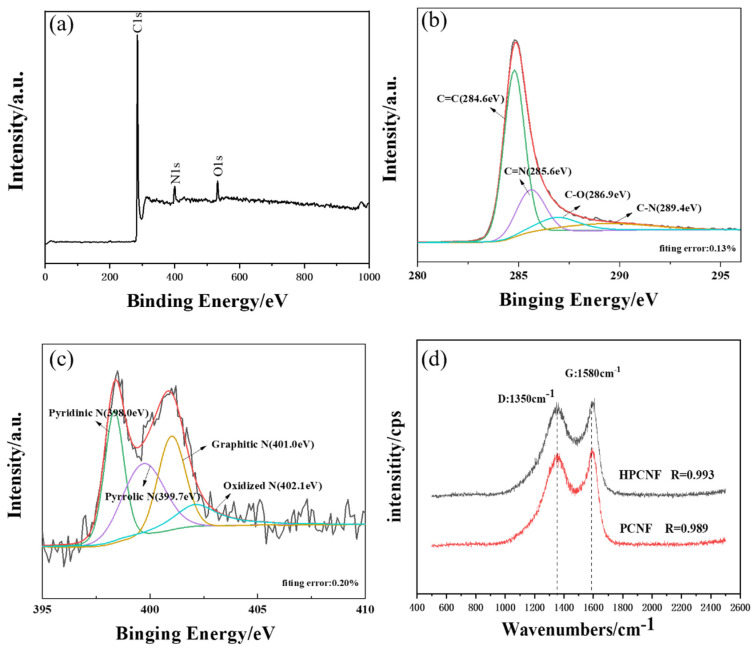
(**a**) XPS spectrum of HPCNF; (**b**) C 1s high−resolution XPS spectrum; (**c**) N 1s high−resolution XPS spectrum; (**d**) Raman spectroscopy of PCNF and HPCNF.

**Figure 3 materials-15-07273-f003:**
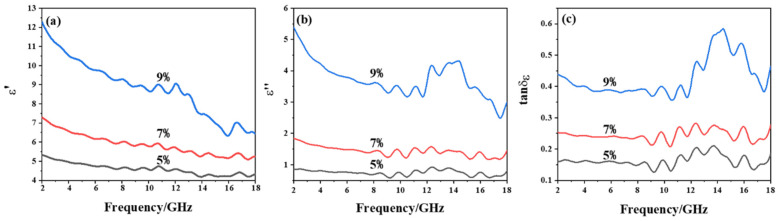
Complex permittivity of PCNF composites: (**a**) real part of complex permittivity; (**b**) imaginary part of complex permittivity; (**c**) loss factor.

**Figure 4 materials-15-07273-f004:**
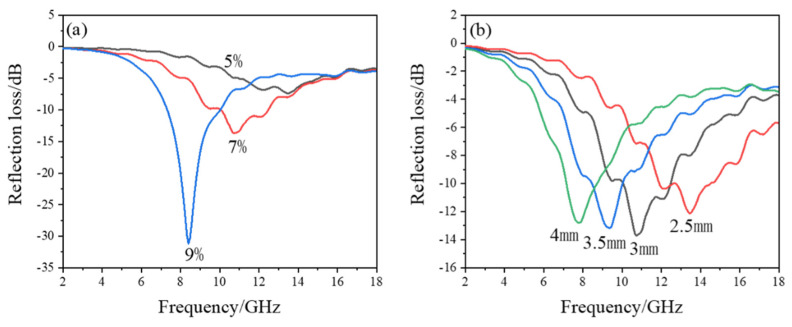
Reflection loss of PCNF composites: (**a**) different contents of PCNF at 3 mm thickness; (**b**) varied thicknesses at PCNF content of 7%.

**Figure 5 materials-15-07273-f005:**
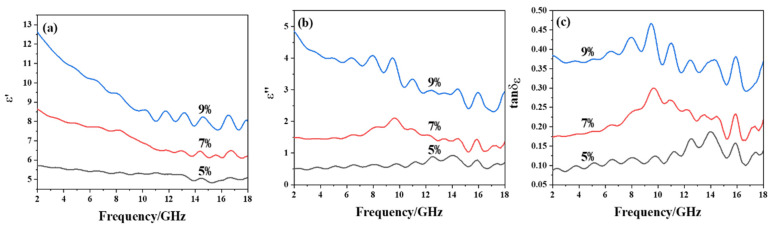
Complex permittivity of HPCNF composites: (**a**) real part; and (**b**) imaginary part of complex permittivity; (**c**) loss factor.

**Figure 6 materials-15-07273-f006:**
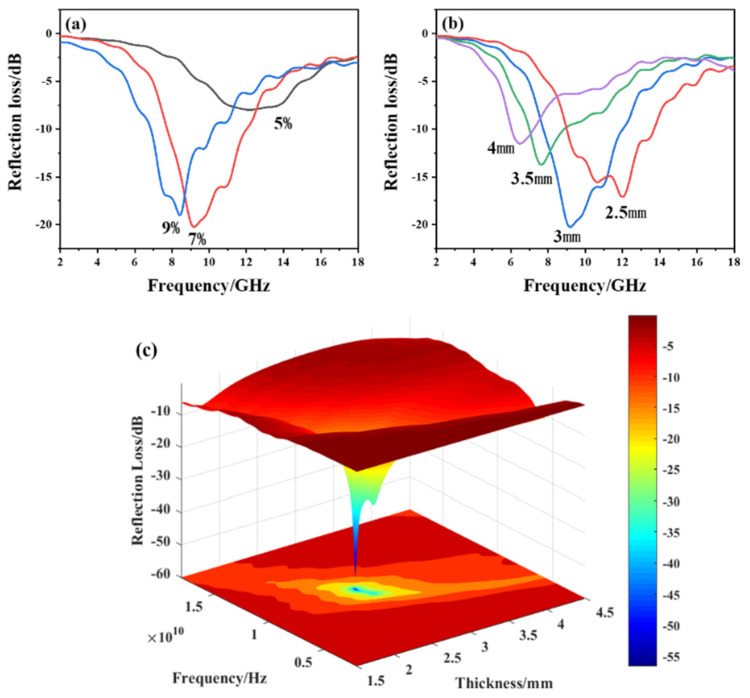
Reflection loss of HPCNF composites: (**a**) different contents of HPCNF at 3 mm thickness; (**b**) varied thicknesses at HPCNF content of 7%; (**c**) relationship between reflection loss and thickness as well as frequency at HPCNF content of 7%.

**Figure 7 materials-15-07273-f007:**
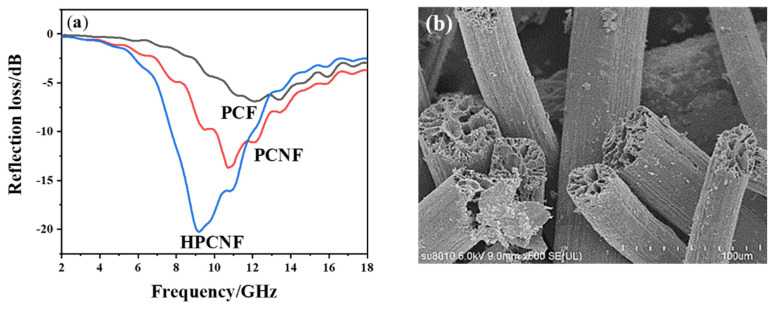
(**a**) The reflection loss of composites filled with different absorbents at the same content of 7%; (**b**) PCF morphology.

**Figure 8 materials-15-07273-f008:**
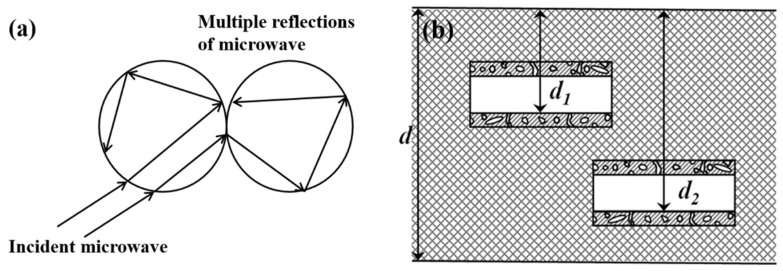
Effect of porous (**a**) and hollow (**b**) structures on microwave transmission and dissipation.

**Figure 9 materials-15-07273-f009:**
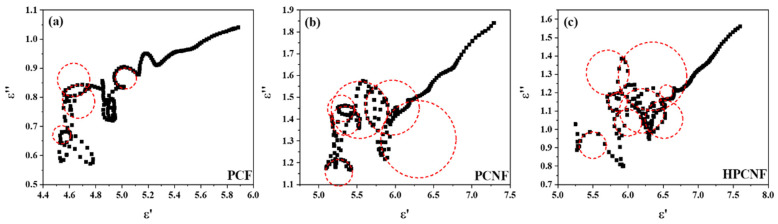
Cole−Cole plots of microwave absorbing composites with different absorbers: (**a**) PCF; (**b**) PCNF; (**c**) HPCNF.

**Figure 10 materials-15-07273-f010:**
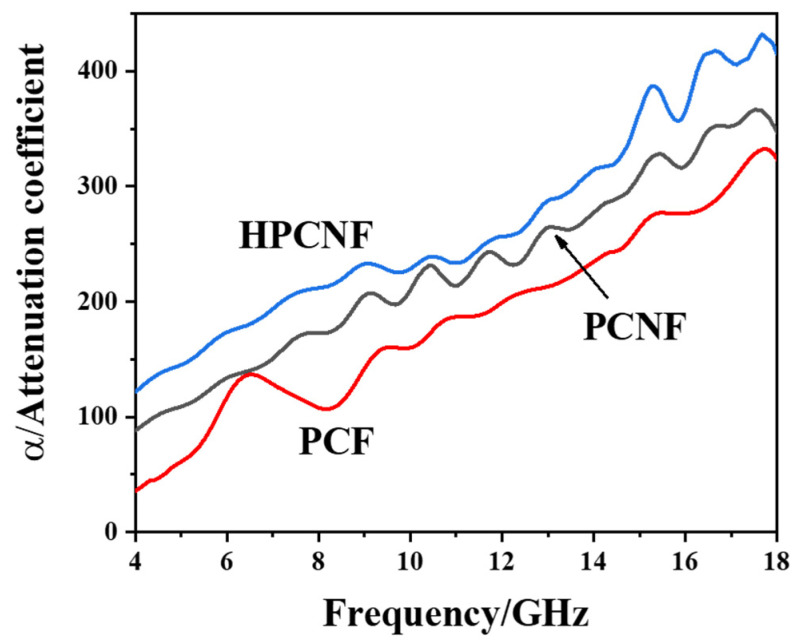
Attenuation coefficients of microwave absorbing composites prepared with different absorbers.

## Data Availability

Not applicable.

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
