# Peer review of "Preparation of Hollow Porous Carbon Nanofibers and Their Performance and Mechanism of Broadband Microwave Absorption"

_materials, 2022, doi:10.3390/ma15207273_

Round 1

Reviewer 1 Report

Shao and colleagues prepared porous carbon nanofiber (PCNF) and hollow porous carbon nanofibers (HPCNF) through (coaxial) electrospinning, preoxidation and carbonization of two precursor fibers. The precursors were prepared using PAN/PMMA blend fibers for PCNF and sheath-core blend fibers with PAN/PMMA as the sheath and PMMA as the core for HPCNF. HPCNF composites show better broadband microwave absorption performance than PCNF and PCF composites due to the high specific surface area and increased number of mesopores. The pores and hollow structure provide more interfaces, increasing the transmission path of electromagnetic waves, and increasing the probability of meeting the interference phase elimination of reflected electromagnetic waves reflected from different interfaces.

Questions and Comments:

1.  There are many small mistakes in the text. Read the manuscript more carefully. English correction is required before publication.

2. The title is grammatically incorrect.

3. In the Introduction section, the novelty of this study should be further emphasized.

4. In section 2.4, how did the authors check whether the coaxial ring sample is uniform? SEM or TEM?

5. In Figures 2 and 4, add the sample thickness.

Author Response

Dear Reviewer,

      Thank you for your letter and your comments on our manuscript entitled “Preparation of Hollow Porous Carbon Nanofiber and their Performance and Mechanism of Broadband Microwave Absorption”. We have made the following changes to the issues you raised.

    1. There are many small mistakes in the text. Read the manuscript more carefully. English correction is required before publication.

    Response: We have made changes to some small mistakes within the text, the results of which you can see in the text.

    2. The title is grammatically incorrect.

    Response: We have made changes to the title.

    3. In the Introduction section, the novelty of this study should be further emphasized.

    Response: We have revised the introduction to further highlight the novelty of preparing the hollow porous carbon nanofiber.

    4. In section 2.4, how did the authors check whether the coaxial ring sample is uniform? SEM or TEM?

    Response: SEM tests have been carried out on the prepared coaxial rings, which you can see in 3.1 of the text.

    5. In Figures 2 and 4, add the sample thickness.

    Response: For your question about adding thickness to the electromagnetic parameters in Figures 2 and 4, we have used coaxial rings to test the electromagnetic parameters of the absorbing samples. The coaxial ring mold has a fixed size, refer to 2.4. The coaxial ring sample size: inner diameter: 3.04 mm, outer diameter: 7 mm, thickness: 2 mm.

    The effect of the thickness of the composite material on the absorbing properties at a fixed content of absorbent, the reflection loss of the material at different thicknesses can be calculated from its electromagnetic parameters with the following equation.

    We hereby resubmit the revised manuscript and hope that all corrections are satisfactory. Please feel free to contact us with any questions and we look forward to your decision.

                                                                                                   Sincerely,

                                                                                                   Guang Li

                                                                                                  2022/9/14

E-mail: lig@dhu.edu.cn

Reviewer 2 Report

In the manuscript entitled "Preparation of Hollow Porous Carbon Nanofiber and their 2 Broadband Microwave Absorption Performance and Mecha-3 nisms", the authors have prepared the porous carbon nanofibers and hollow porous carbon nanofibers by the preoxidation and carbonization of the PAN/PMMA blend fibers. The prepared materials are represented by the authors to be used as microwave absorbing composites.

 Although the study is quite interesting and analysis of microwave absorption performance was described in detail, I would be able to recommend this manuscript for further publication only after major revision. Please, find below comments.

 1)     “Previous researches demonstrated that the presence of pores in carbon fibers…” There are no any references on previous researches. Please add some.

2)     The prepared porous carbon nanofibers were characterized not enough.

– There is no data about the specific surface area.

– There is no structural analysis of carbon nanofibers, such as XPS or Raman spectroscopy.

Author Response

Dear Reviewer,

     Thank you for your letter and your comments on our manuscript entitled “Preparation of Hollow Porous Carbon Nanofiber and their Performance and Mechanism of Broadband Microwave Absorption”. We have made the following changes to the issues you raised.

    1. “Previous researches demonstrated that the presence of pores in carbon fibers…” There are no any references on previous researches. Please add some.

    Response: We have included the corresponding references.

    2. The prepared porous carbon nanofibers were characterized not enough.

    –There is no data about the specific surface area.

    –There is no structural analysis of carbon nanofibers, such as XPS or Raman spectroscopy.

    Response:

    –Data for the specific surface area can be found in 3.1 of the text.

    –We have characterized porous carbon nanofibers, including XPS and Raman spectroscopy, which you can see in 3.2 of the text.

    We hereby resubmit the revised manuscript and hope that all corrections are satisfactory. Please feel free to contact us with any questions and we look forward to your decision.

                                                                                                         Sincerely,

                                                                                                          Guang Li

                                                                                                          2022/9/14

E-mail: lig@dhu.edu.cn

Reviewer 3 Report

The manuscript written by Shao et al. “Preparation of Hollow Porous Carbon Nanofiber and their Broadband Microwave Absorption Performance and Mechanisms” is a well discussed with details on the microwave absorption studies. 

The following points can be addressed 

  1. To better characterize the carbon material FTIR and Raman spectra can be added. 

  1. Some relevant references that can be added are Electrochimica Acta 2019, 313, 341-351, 10.1039/C9RA02431J

Author Response

Dear Reviewer,

    Thank you for your letter and your comments on our manuscript entitled “Preparation of Hollow Porous Carbon Nanofiber and their Performance and Mechanism of Broadband Microwave Absorption”. We have made the following changes to the issues you raised.

    1. To better characterize the carbon material FTIR and Raman spectra can be added.

    Response: The analysis of the Raman spectra of PCNF and HPCNF can be found in 3.2 of the text. The test of FTIR you mentioned has not been added, because the fibers are carbonised and have a graphitised structure, mostly in the form of C=C bonding, with almost no specific functional groups appearing and no specific absorption peaks appearing.

    2. Some relevant references that can be added are Electrochimica Acta 2019, 313, 341-351, 10.1039/C9RA02431J

    Response: We have read the reference you refer to, which is mainly about electrochemistry. It is not closely related to the article we are submitting, so please understand that it is not used as a reference.

    We hereby resubmit the revised manuscript and hope that all corrections are satisfactory. Please feel free to contact us with any questions and we look forward to your decision.

Sincerely,

Guang Li

2022/9/14

Email: lig@dhu.edu.cn

Round 2

Reviewer 2 Report

I am satisfied with the answers om my previous comments but the interpretation of the results of the XPS analysis raises questions. In Fig. 2b and c, the authors drew the background line very carelessly. The authors should bring all drawings to the same type of background line (Shirley) and remake the peaks fitting because the area of these peaks raises questions. The drawings should show raw spectrum, fitted peaks, summed spectrum, and peak fitting error.

Author Response

Dear Reviewer,

    Thank you for your letter and your comments on our manuscript entitled “Preparation of Hollow Porous Carbon Nanofiber and their Performance and Mechanism of Broadband Microwave Absorption”. We have made the following changes to the issues you raised.

1. I am satisfied with the answers om my previous comments but the interpretation of the results of the XPS analysis raises questions. In Fig. 2b and c, the authors drew the background line very carelessly. The authors should bring all drawings to the same type of background line (Shirley) and remake the peaks fitting because the area of these peaks raises questions. The drawings should show raw spectrum, fitted peaks, summed spectrum, and peak fitting error.

Response: We redrew Figure 2 (b) and (c), with Shirley used for all background line types, and remade the peaks fitting. The raw spectrum, fitted peaks, summed spectrum, and peak fitting error can be seen in Figure 2 (b) and (c).

    We hereby resubmit the revised manuscript and hope that all corrections are satisfactory. Please feel free to contact us with any questions and we look forward to your decision.

Sincerely,

Guang Li

2022/9/22

Reviewer 3 Report

The manuscript can be accepted in its present form.

Author Response

Dear Reviewer,
     Thank you for your letter and for your acknowledgement of our manuscript entitled "Preparation of Hollow Porous Carbon Nanofiber and their Performance and Mechanism of Broadband Microwave Absorption".

     Please feel free to contact us with any questions 

Sincerely,

Guang Li

2022/9/22